

# *HaMADS3*, *HaMADS7*, and *HaMADS8* are involved in petal prolongation and floret symmetry establishment in sunflower (*Helianthus annuus* L.)

Qian Wang, Zhou Su, Jing Chen, Weiying Chen, Zhuoyuan He, Shuhong Wei, Jun Yang and Jian Zou

College of Life Sciences, China West Normal University, Nanchong, Sichuan, China

## ABSTRACT

The development of floral organs, crucial for the establishment of floral symmetry and morphology in higher plants, is regulated by MADS-box genes. In sunflower, the capitulum is comprised of ray and disc florets with various floral organs. In the sunflower *long petal mutant* (*lpm*), the abnormal disc (ray-like) floret possesses prolongated petals and degenerated stamens, resulting in a transformation from zygomorphic to actinomorphic symmetry. In this study, we investigated the effect of MADS-box genes on floral organs, particularly on petals, using WT and *lpm* plants as materials. Based on our RNA-seq data, 29 MADS-box candidate genes were identified, and their roles on floral organ development, especially in petals, were explored, by analyzing the expression levels in various tissues in WT and *lpm* plants through RNA-sequencing and qPCR. The results suggested that *HaMADS3*, *HaMADS7*, and *HaMADS8* could regulate petal development in sunflower. High levels of *HaMADS3* that relieved the inhibition of cell proliferation, together with low levels of *HaMADS7* and *HaMADS8*, promoted petal prolongation and maintained the morphology of ray florets. In contrast, low levels of *HaMADS3* and high levels of *HaMADS7* and *HaMADS8* repressed petal extension and maintained the morphology of disc florets. Their coordination may contribute to the differentiation of disc and ray florets in sunflower and maintain the balance between attracting pollinators and producing offspring. Meanwhile, Pearson correlation analysis between petal length and expression levels of MADS-box genes further indicated their involvement in petal prolongation. Additionally, the analysis of *cis*-acting elements indicated that these three MADS-box genes may regulate petal development and floral symmetry establishment by regulating the expression activity of *HaCYC2c*. Our findings can provide some new understanding of the molecular regulatory network of petal development and floral morphology formation, as well as the differentiation of disc and ray florets in sunflower.

Corresponding authors
Jun Yang, yangjun@cwnu.edu.cn
Jian Zou, zoujian@cwnu.edu.cn

## INTRODUCTION

In the reproductive system of angiosperms, the flower is the basis of sexual reproduction and consequently, the most important guarantee of progeny production in higher plants. Flower development initiation is a key event for the transformation from vegetative to reproductive growth (*Celedón-Neghme, Gonzáles & Gianoli, 2007*). Typically, flower development originates from floral induction, and shoot apical meristems can differentiate into inflorescence meristems, then transform into floral meristems (FMs), and trigger floral organogenesis (*Denay et al., 2017*; *Wagner, 2017*). Since the normal development of floral organs determines the type, position, number, and size of floral organs, it can affect the reproductive capacity of plants (*Fan et al., 2022*). Furthermore, floral organ development may determine the external morphology of flowers (*Shan et al., 2019*). Notably, floral symmetry is closely related to the morphogenesis of floral organs. Based on floral symmetrical features, flowers can be divided into three types: asymmetric flowers with no plane of symmetry, zygomorphic flowers with only one plane of symmetry, and actinomorphic flowers with multiple planes of symmetry (*Chai et al., 2023*). Previous studies have shown that zygomorphic flowers evolved from actinomorphic types and possess a stronger ability to recruit pollinators, giving them a stronger reproductive ability and evolutionary competitiveness (*Fernández-Mazuecos & Glover, 2017*; *Moyroud & Glover, 2017*; *Shen et al., 2021*).

The establishment of floral symmetry is closely related to floral organ morphogenesis, such as the shape and size of floral organs (*Hileman, 2014*). In some plants, floral organs that are located in the same whorl develop synchronously through equilateral development, resulting in the formation of actinomorphic flowers, such as disc floret in Asteraceae (*He et al., 2022*). Additionally, the asynchronous development of floral organs depended on non-equilateral development contributes to the formation of zygomorphic or asymmetric flowers (*Almeida & Galego, 2004*). In most plants of the Papilionoideae subfamily, the petal primordia develop asynchronously, forming a vexillum petal, two wing petals, and two keel petals, leading to the formation of a zygomorphic flower (*Tucker, 2003*). For *Chrysanthemum rhombifolium*, the dorsal, lateral, and ventral petal primordia can develop differentially and then fuse to form zygomorphic florets (*Shen et al., 2021*). Interestingly, the petals of ray florets are also zygomorphic in sunflower.

Floral organ development and morphogenesis are regulated by internal and external factors (including floral organ identity genes) in order to accurately control the specific morphology, position, number, and size of floral organs. Currently, the mechanism of floral organ development has been investigated extensively in *Arabidopsis thaliana*, *Antirrhinum majus*, and other flowering plants, and the classical ABCDE model for floral development has been established (*Alvarez-Buylla et al., 2010*). In this model, A-class genes determine the formation of sepals and petals, B-class genes control petals and stamens, C-class genes regulate stamens and pistils, and E-class genes are involved in the development of all flower organs (*Theißen, Melzer & Rümpler, 2016*; *Patil et al., 2023*). As members of the ABCDE model, MADS-box genes function in determining floral organ identity, and act in the occurrence and development of floral organs (*Ng & Yanofsky, 2001*;

*Parenicova et al., 2003*; *Wei et al., 2015*). In *A. thaliana*, the function deficiency of these MADS-box genes can result in uncertainty regarding the floral organ's identity, leading to an abnormal morphogenesis, although they can be partially recovered by genetic complementation (*Lamb & Irish, 2003*; *Jack, 2004*; *Shchennikova et al., 2004*).

It has been reported that MADS-box genes play an important role during the development of floral organs that are related to the establishment of floral symmetry. In *Gerbera hybrida*, the expression levels of numerous MADS-box genes exhibited significant differences between ray and disc floret, including *GERBERA REGULATOR OF CAPITULUM DEVELOPMENT 1* (*GRCD1*), *GRCD2*, *GRCD3*, *GRCD5*, and *G. hybrida SUPPRESSOR OF OVEREXPRESSION OF CONSTANS 1*-like (*GhSOC1*), suggesting their effect on the establishment of flower symmetry (*Kotilainen et al., 2000*; *Uimari et al., 2004*; *Laitinen et al., 2006*). In *Oncidium* Gower Ramsey, when the expression level of *OMADS5* was down-regulated, the sepals and petals converted into lip-like sepals and lip-like petals, respectively, and the standard zygomorphic flowers then transformed into asymmetric ones (*Chang et al., 2010*). Additionally, MADS-box genes modulate the determinacy of floral symmetry by controlling the identity, shape, and size of floral organs, which necessitates interaction with other genes related to floral development (*Lucibelli, Valoroso & Aceto, 2020*). In *Medicago truncatula*, the MADS-box genes *AGAMOUS* (*AG*) homologs, *MtAGa*, and *MtAGb* regulated floral symmetry by modulating the expression of *CYCLOIDEA*-like (*CYC*) genes, *MtCYC1*, *MtCYC2*, and *MtCYC3*. In the double mutant *mtaga/mtagb-2*, the stamens and carpels were transformed into flag-like petals, and the dorsal petals developed abnormally due to the ectopic expression of *MtCYC* caused by the function loss of *MtAGa* and *MtAGb* (*Zhu et al., 2018*).

As the typical representative of Asteraceae, the sunflower (*Helianthus annus* L.) is a model species for investigating the development and regulation of floral symmetry because of their capitulum comprised of zygomorphic ray florets and actinomorphic disc florets. To investigate the function of related genes during the process of sunflower floral symmetry determinacy would shed light on the complex regulatory network of this process in angiosperms. However, the role of MADS-box genes on the morphogenesis of floral organs and establishment of floral symmetry is still unclear in sunflower. In this study, *long petal mutant* (*lpm*) plants were used as experimental materials to explore the function of MADS-box genes on the development of floral organs, especially on petals, and the establishment of floral symmetry in sunflower. It was found that high levels of *HaMADS3* (LOC110941381) and low levels of *HaMADS7* (LOC110865152) and *HaMADS8* (LOC110894044) promoted petal prolongation and maintained the morphology of ray florets, while low levels of *HaMADS3* and high levels of *HaMADS7* and *HaMADS8* repressed petal extension and maintained the morphology of disc florets; therefore they are involved in the morphogenesis and symmetry establishment of florets in sunflower. Our findings help to reveal the roles of MADS-box genes in petal prolongation and floral symmetry establishment, as well as provide some new clues for the molecular regulatory network of morphogenesis and symmetries in sunflower.

## MATERIALS AND METHODS

### Plant materials

The sunflower GuangWu Mountain Wild Type (WT) was treated by space mutation *via* "Shenzhou No.4" (*Wu et al., 2020*). Space mutation is an efficient technology used for changing the genetic characteristics of living organisms through pace-related factors, such as space radiation, space microgravity, cosmic irradiation, and space magnetic fields. The *lpm* plants were screened from the offspring of space mutants after several years of inbreeding. These *lpm* plants were divided into five types: type V, IV, III, II, and I. From type V to I, the mutant traits gradually became obvious. In type V plants, the phenotype was similar to that of WT plants. The type I plants possessed the extreme phenotype, with prolongated petals and degenerated stamens in the most disc floret (*He et al., 2022*). In our experiment, only the type I *lpm* plants were used for analyzing the effect of MADS-box genes on petal prolongation and floret symmetry establishment. The experimental materials were planted in the field of the College of Life Sciences, China West Normal University (Nanchong, Sichuan Province, P. R. China, 30°49′N, 106°4′E). All plants were sown in March and collected in July and August 2022 under natural conditions (an average temperature of 25 ± 5 °C and average rainfall 85.8 mm). Fertilization and weeding were performed once a month and the plants were irrigated twice a month.

### Morphological and histological analysis

When blooming, the floret materials from WT and *lpm*, including ray and disc florets, were harvested to compare their morphogenesis, and the floret petals of WT and *lpm* plants (within the $1^{st}$ (ray floret), $5^{th}$, $15^{th}$, and $19^{th}$ parastichy) were collected to analyze the petal length using a vernier caliper with 20 biological repeats. The petal length was determined by measuring the petal naturally straightened on a flat surface from bottom to top. Subsequently, the disc florets at different floral developmental stages (25 days prior to blooming (DPB), 15, 5, and 0 DPB) were harvested to analyze the morphological differences of disc florets, stamens, and pistils between WT and *lpm via* stereomicroscope (Olympus, Tokyo, Japan). The disc florets (at 30, 25, 20, and 15 DPB) were collected and fixed in FAA solution (70% ethanol: acetic acid: formaldehyde, 90:5:5, v/v) for histological analysis. Subsequently, all samples were dehydrated in graded ethanol (30%, 50%, 70%, 80%, 90%, 100%, and 100% ethanol), each for 1 h at room temperature. The dehydrated materials were then incubated in the ethanol with xylene as follows: ethanol/xylene (1:3, v/v) for 2 h, ethanol/xylene (2:3, v/v) for 2 h, ethanol/xylene (1:1, v/v) for 2 h, three changes of xylene for 1 h. The materials were then set in xylene/paraffin (1:1, v/v) for 12–20 h at 40 °C and in paraffin at 60 °C with three changes for 1 h as interval, and then embedded in pure molten wax. Samples were sectioned into pieces with 7 μm using microtome (Leica, Wetzlar, Germany). The sections were deparaffinized 10 min in xylene two times and rehydrated as follows: 100%, 100%, 95%, 80%, and 70% ethanol and distilled water, each for 5 min. Sections were then stained with toluidine Blue O (G3668, Solarbio, Beijing, China) for 5 min and washed in water. After that, sections were dehydrated for 1 min through the ethanol series (70%, 80%, 95%, 100%, and 100% ethanol) and set in

xylene for 15 min. Finally, the sections were covered with glass cover and observed under optical microscope (Olympus, Tokyo, Japan). Petals from different positions (the 1st, 5th, 15th, and 19th parastichy) on the capitulum were selected to analyze the cells located in the upper, middle, and lower part of the petal in order to record cell size and number data.

## RNA-sequencing and data analysis

The total RNA of root, stem, leaf, and flower were extracted using an E.Z.N.A®Plant RNA Kit (R6827; Omega, Norcross, GA, USA) and sent to Shanghai Majorbio Biopharm Technology Co., Ltd. (Shanghai, China) for RNA-sequencing. The cDNA library was constructed using the Illumina NovaSeq Reagent Kit (Illumina, San Diego, CA, USA) and 1 µg of RNA, and RNA-sequencing was performed through the Illumina NovaSeq 6000 platform. The raw paired end reads were trimmed, and the quality of the raw paired end reads was controlled by fastp (*Chen et al., 2018*). The clean reads were aligned to the sunflower genome (https://www.ncbi.nlm.nih.gov/genome/?term=txid4232[orgn]) using HISAT2 software (*Kim, Langmead & Salzberg, 2015*), and the mapped reads of each sample were assembled *via* StringTie (*Kovaka et al., 2019*). The number of raw and clean reads, percentage of total-mapped reads, and percentage of unique-mapped reads obtained from each sample are listed in Table S1. The expression levels of the transcripts were calculated using the transcripts per million reads (TPM) method. The genes were annotated through web sites, including GenBank of National Center for Biotechnology Information (NCBI) (https://www.ncbi.nlm.nih.gov/genbank/) and the European Molecular Biology Laboratory (EMBL) (https://www.embl.de/).

## The identification of MADS-box genes and analysis of conserved motifs and phylogenetic tree

The putative MADS-box genes were identified based on the sunflower genome (https://www.ncbi.nlm.nih.gov/genome/?term=txid4232[orgn]) and our RNA-seq data from a collection of root, stem, leaf, and flower tissues. Using all MADS-box protein sequences of *A. thaliana* obtained from the TAIR database (http://www.arabidopsis.org) as queries, the MADS-box sequences of sunflower were identified through BLASTP. Subsequently, based on the annotations in our RNA-seq data, the members of the MADS-box family were finally identified and analyzed in this study. Notably, the MADS-box members that could be effectively detected in our RNA-seq data were identified, but not all the members of the MADS-box family in sunflower. The detailed information of these MADS-box genes is provided in Tables S2–S4, including transcript sequences, detailed annotations, and TPMs.

The protein sequences of the 11 reported MADS-box genes in sunflower (*Dezar et al., 2003*; *Shulga et al., 2008*) were then aligned with all candidate MADS-box genes in our study using DNAMAN 8.0 (https://dnaman.software.informer.com/) and annotated in Table S3. Subsequently, the conserved motifs of protein sequences encoded by the candidate genes were analyzed and visualized *via* MEME (http://memesuite.org/tools/meme). The multiple sequence alignment was implemented by Clustal 2.0 (http://www.clustal.org/) using the protein sequences of the MADS-box genes of sunflower and *A. thaliana* (see NCBI IDs and TAIR IDs in Table S5). A phylogenetic tree was constructed

using MEGA 11.0.13 (*Tamura, Stecher & Kumar, 2021*), using neighbor-joining (NJ) with 1,000 bootstrap replications, *p*-distance model, and the pairwise deletion of gaps.

## RNA extraction and qPCR analysis

The roots, stems, leaves, and flowers from WT were collected for analyzing the tissue expression pattern of the MADS-box genes. The disc florets of WT and *lpm* at different stages were collected to explore the temporal expression pattern of MADS-box genes. The petals, pistils, and bracts in the WT disc florets and abnormal disc (ray-like) florets of *lpm* were used to investigate the spatial expression patterns of these genes. The petals from ray and disc florets in WT were applied to explore the function of candidate MADS-box genes during the process of petal development, and the floret petals of WT and *lpm* plants within the 1st, 5th, 15th, and 19th parastichy were collected to analyze the expression levels of MADS-box genes in petals across different positions. The detailed information of materials used for quantitative real-time PCR (qPCR) are listed in Table S6. All materials used in this study for qPCR were gathered from 15 individual plants that were grown in the same conditions and then mixed for RNA extraction. At least 0.5 g of materials were harvested, immediately frozen in liquid nitrogen, and then stored at −80 °C for no more than 3 months.

Total RNA was extracted using the E.Z.N.A®Plant RNA Kit. NanoDrop2000 (ThermoFisher, New York, NY, USA) was used to detect quantification, OD260/280, OD260/230, and contamination assessment of RNA. Then 1% agarose gel electrophoresis was used to analyze RNA integrity *via* electrophoresis apparatus (Bio-Rad, Hercules, CA, USA) and gel imaging system (Bio-Rad, Hercules, CA, USA), using Tris-Borate-EDTA (BL540A, Biosharp, Hefei, Anhui, China) as buffer and SuperRed (BS354A, Biosharp, Hefei, Anhui, China) as dye. See detailed results in Table S7 and Fig. S1.

The cDNA was obtained using the PrimeScript™ RT reagent Kit with gDNA Eraser (RR047A, TaKaRa, Shiga, Japan). The reaction for removing gDNA was performed using 10 μL mix including: 2.0 μL 5 × gDNA eraser buffer, 1.0 μL gDNA eraser, 1.0 μg total RNA, and adding RNase free $dH_2O$ up to 10 μL. The 10 μL mix was then incubated at 42 °C for 10 min. RNA reverse transcription was performed using 20 μL mix including: 10 μL mix after removing gDNA reaction, 1.0 μL PrimeScript RT enzyme mix I, 1.0 μL RT primer mix, 4.0 μL 5×PrimeScript buffer 2, and 4.0 μL RNase free $dH_2O$. The 20 μL mix was incubated at 37 °C for 15 min and 85 °C for 5 s. Finally, the 20 μL cDNA solution was diluted 10 times and stored at −20 °C.

qPCR was performed using TB Green® Premix Ex Taq™ II (RR820A, TaKaRa, Shiga, Japan) *via* CFX96™ Real-Time System (Bio-Rad, Hercules, CA, USA). qPCR reaction was performed using 20 μL mix including: 10 μL TB Green Premix Ex Taq II (Tli RnaseH Plus), 1.0 μL forward primer (10 μM), 1.0 μL reverse primer (10 μM), 7.0 μL Rnase free $dH_2O$, and 1.0 μL cDNA. qPCR was performed with following program: 30 s at 95 °C, then 40 cycles of 5 s at 95 °C, and 30 s at 56 °C. Data analysis for qPCR was performed by Bio-Rad CFX maestro 1.1 software (version 4.1.2433.1219, https://www.bio-rad.com/). For each sample, 15 biological (pooled) and three technical replicates were conducted. Gene expression levels were calculated using the $2^{-\Delta\Delta Ct}$ method, with *elongation factor 1-*

*alpha* (*EF-1α*; GenBank accession: XM_022117715.2; also known as *EF1A*) as the reference gene (*De Paolis et al., 2022*; *He et al., 2022*; *Najafabadi & Amirbakhtiar, 2023*). All primers are listed in Table S8. The MIQE checklist for qPCR is listed in Table S9.

### Tissue expression pattern and correlation analysis

The tissue expression pattern was analyzed based on the TPMs in our RNA-seq data (Table S4) and visualized *via* the pheatmap package of R 4.3.1 (*R Core Team, 2023*). The relationship between petal length and expression levels of *HaMADS3*, *HaMADS7*, and *HaMADS8* were analyzed using the Pearson correlation coefficient (PCC). Pearson correlation analysis was performed *via* R WGCNA package. PCC > 0.6 and *P*-value < 0.05, or PCC < −0.6 and *P*-value < 0.05 were considered positively or negatively correlated, respectively (*Zhang et al., 2015*).

### Analysis of *cis*-acting elements

The 2,000 bp sequences (−2,000 bp) upstream of the transcription start site of *HaCYC2c*, *HaMADS3*, *HaMADS7*, and *HaMADS8* were obtained from NCBI (https://www.ncbi.nlm. nih.gov/). The *cis*-acting elements in the promoter region of *HaCYC2c* bound by MADS-box transcriptional factors, as well as in the promoter region of *HaMADS3*, *HaMADS7*, and *HaMADS8* bound by HaCYC2c, were analyzed by JASPAR (*Khan et al., 2018*). The results were visualized using TBtools (*Chen et al., 2020*).

### Statistical analysis

Statistical analysis was performed by one-way ANOVA and Tukey multiple comparison using SPSS 22.0 (IBM, Armonk, NY, USA).

## RESULTS

### Morphogenesis of florets in WT and *lpm*

To analyze the morphogenetic difference between WT and *lpm*, the capitulum structure and composition, floret symmetry, and morphogenesis of floral organs were investigated. The results showed that the capitulum of WT was composed of multiple whorls of actinomorphic disc florets with shorter petals surrounded by a single whorl of zygomorphic ray florets with longer petals (Fig. 1A). In WT plants, the zygomorphic ray floret consists of zygomorphic petals and a pseudo-ovary, while the actinomorphic disc floret possesses a single bract, a pair of pappus, one actinomorphic corolla fused by five petals, five synantherous stamens, and one pistil (Figs. 1B and 1C). In *lpm*, an insignificant difference was observed in ray florets, compared with WT (Fig. 1C). Comparatively, in *lpm*, the petals of abnormal disc florets were significantly elongated and the stamens were partially degenerated, while the structure of other floral organs remained normal. The symmetry of disc florets in *lpm* was transformed from actinomorphic to zygomorphic, their outline was similar to that of ray florets in WT, and these florets were thus defined as ray-like florets (Figs. 1A–1C). Notably, the ray-like florets in *lpm* were not completely uniform in morphology, and the transformation of petal length and symmetry gradually weakened from outside to center (Fig. 1D).

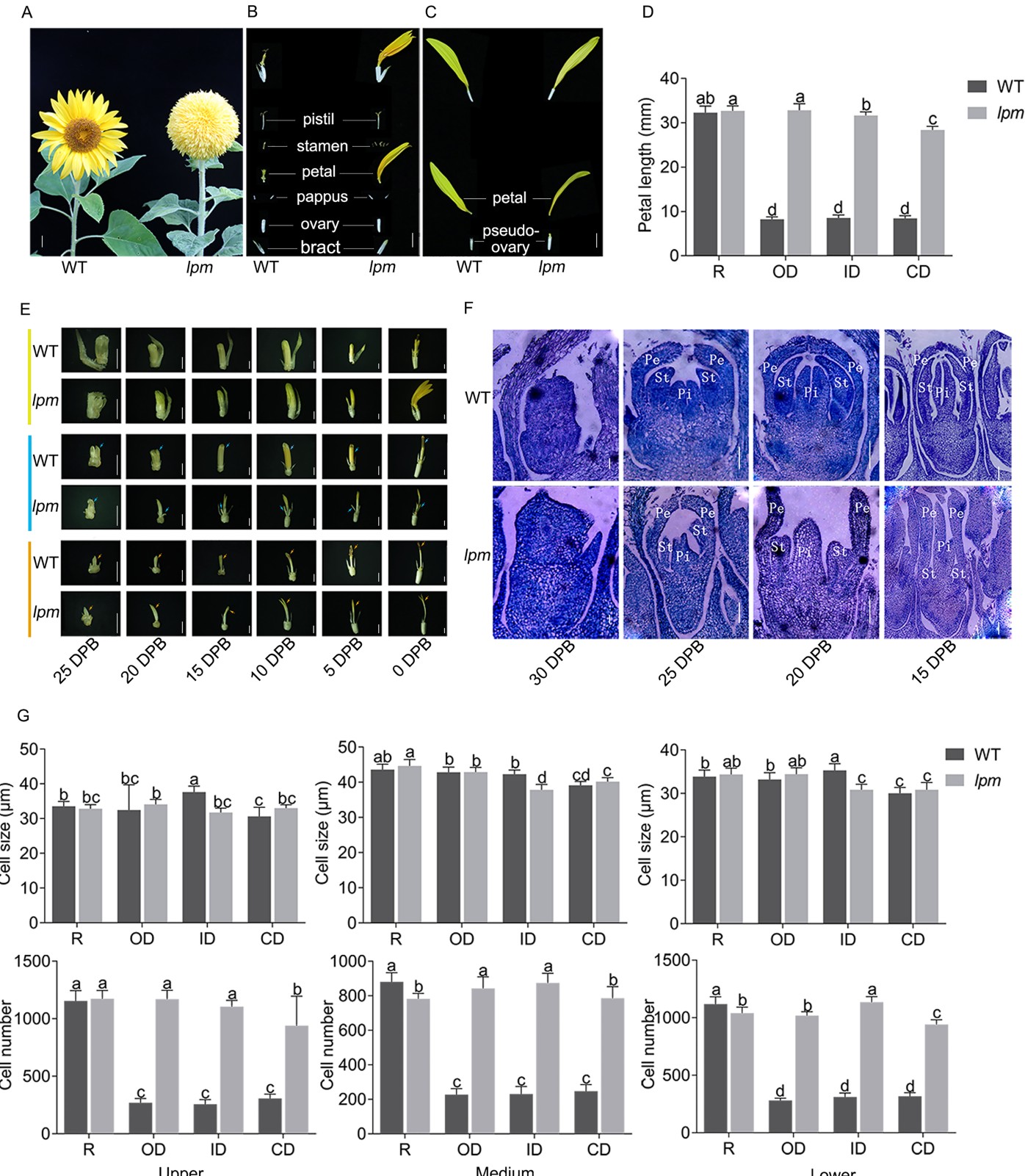

**Figure 1 Morphological and histological comparison between WT and *lpm*.** (A) Capitulum. Bar = 1 cm. (B and C) Anatomy of disc and ray floret. Bar = 5 mm. (D) Petal length. R: ray floret (the 1st parastichy). OD: outer disc floret (the 5th parastichy). ID: intermediate disc floret (the 15th parastichy). CD: central disc floret (the 19th parastichy). Each column represents the mean ± SD of 20 biological replicates. Different lowercase letters

**Figure 1 (continued)**
represent significant differences (*P* < 0.05 by one-way ANOVA analysis and Tukey multiple comparison). (E) Morphological analysis of disc floret (yellow), stamen (blue), and pistil (orange) at different stages. DPB: days prior to floret blooming, corresponding to the reproductive stage. 25, 20, 15, and 10 DPB: bar = 1 mm. 5 and 0 DPB: bar = 2 mm. (F) Histological analysis of disc floret at different stages. Pe: petal. St: stamen. Pi: pistil. 30 and 25 DPB: bar = 0.1 mm. 20 and 15 DPB: bar = 0.2 mm. (G) Cell size and number analysis of petal. Each column represents the mean ± SD of 20 biological replicates. Different lowercase letters represent significant differences (*P* < 0.05 by one-way ANOVA analysis and Tukey multiple comparison). 

Analyzing the development schedule of floral organs of disc florets *via* the morphological and histological method indicated that the difference between WT and *lpm* at 25 DPB was insignificant, while the divergency initially emerged at 20 DPB. In particular, the petals in *lpm* started to elongate faster than those of the WT plants, and the discrepancy of petal length gradually increased afterwards (Fig. 1E). Coincidentally, until 25 DPB, stamen development was almost synchronized between *lpm* and WT, indicating that they shared similar stamen morphology. Subsequently, stamens developed following the normal schedule in WT, while stamen development in *lpm* was suppressed, causing only stamen traces to be present in the disc florets when blooming (Figs. 1E and 1F). Regarding pistils, an insignificant difference was observed between WT and *lpm* except in color. The pistil stigma was light yellow in disc florets of WT, then gradually darkened at 10 DPB, and finally transformed into purple-red, but the stigma color remained light yellow in *lpm* (Fig. 1E).

To determine the cause of petal prolongation in *lpm*, the cell size and number were analyzed in petals. The results showed an insignificant difference in the cell size presented between WT and *lpm* (Fig. 1G). The cell number of petals in disc florets of *lpm* was significantly higher than that of WT, despite no obvious difference in ray floret petals between WT and *lpm* (Fig. 1G). These results revealed that the increase in cell number was the main contribution to petal prolongation of disc florets in *lpm*.

## Identification of MADS-box genes in sunflower

Based on our RNA-seq data and sunflower genome (https://www.ncbi.nlm.nih.gov/genome/?term=txid4232[orgn]), 29 MADS-box candidate genes were identified: 11 previously reported genes and 18 unreported genes (*Dezar et al., 2003*; *Shulga et al., 2008*). They were labeled *HaMADS1* to *HaMADS29* (see Table S3 for details).

In order to classify the obtained MADS-box genes, 13 conserved motifs of the proteins encoded by these genes were further analyzed (Table 1). The results showed that among these genes, *HaMADS28* and *HaMADS29* belonged to the Type I class because their proteins only contained motifs 1 and 3, while the remaining 27 genes belonged to the Type II class because their proteins contained motifs 1, 2, 3, and 5 simultaneously (Fig. 2A). The results of the phylogenetic analysis indicated that these 29 MADS-box genes were distributed in 11 clades, and the detailed information is listed in Fig. 2B.

## Identifying the MADS-box genes with high expression in flower

To predict the function of the 29 MADS-box genes, their tissue expression patterns were analyzed based on our RNA-seq data (Table S4). Nineteen genes presented their maximum

**Table 1 The sequence of predicted conserved motifs.**

| Motif | Sequence | Length |
|---|---|---|
| 1 | MGRGKIZJKRIENKTNRQVTFSKRRNGLLKKAYELSVLCDAEVALIVFSS | 50 |
| 2 | LLGEDLESLNJKELQNLEKQLETALKRIRERK | 32 |
| 3 | GKLYEYSSSSSMETI | 15 |
| 4 | WTQEYSKLKAKIEVLQRNQRH | 21 |
| 5 | NQLLLEEIEELQKKERELMEZNKALRKKJ | 29 |
| 6 | TCKKKVRSAQDVYKKLMHEFDIRGEDPQYGMIEDAGEYEALYGYPPHIA | 49 |
| 7 | TIPPNDQGZSHNYVDHDVCFCLPGPPPED | 29 |
| 8 | ERYERYSY | 8 |
| 9 | IPKIMRKREQVLEEENKHLMYLVQQSEMAAMGDYQQDEPFSFRVQPMQPN | 50 |
| 10 | PHILTLRLQPNHPNNLHAFPS | 21 |
| 11 | QQHMSLMPGSSGYDDLGPHQPFDGRNDLQVNELQPNNNYSCQDQTPLQLV | 50 |
| 12 | AQAHAISKGMIPPWLYRHIN | 20 |
| 13 | DPPSTGSVAEANAQF | 15 |

expression levels only in flower: *HaMADS1*, *HaMADS3*, *HaMADS7*, *HaMADS8*, *HaMADS9*, *HaMADS11*, *HaMADS12*, *HaMADS15*, *HaMADS16*, *HaMADS18*, *HaMADS19*, *HaMADS20*, *HaMADS21*, *HaMADS22*, *HaMADS23*, *HaMADS24*, *HaMADS25*, *HaMADS26*, and *HaMADS27* (Fig. 3A). The other 10 genes that had high expression levels both in flower and other organs, or were hardly expressed in flower, were set aside for further investigation (Fig. 3A). Therefore, only 19 MADS-box genes were considered to play important roles in flower development.

## Differential expression of MADS-box genes between WT and *lpm* during flower development

To analyze the precise action period of the 19 MADS-box genes during flower development, their expressions were analyzed at four developmental stages in WT and *lpm* using qPCR. In WT sunflower, 11 genes exhibited higher expression at 25 DPB than other stages: *HaMADS7*, *HaMADS8*, *HaMADS9*, *HaMADS11*, *HaMADS12*, *HaMADS15*, *HaMADS21*, *HaMADS23*, *HaMADS24*, *HaMADS26*, and *HaMADS27*. Among these genes, seven (*HaMADS9*, *HaMADS11*, *HaMADS15*, *HaMADS23*, *HaMADS24*, *HaMADS26*, and *HaMADS27*) were hardly expressed or only presented with low levels in the other stages (Fig. 3B). The other eight genes (*HaMADS1*, *HaMADS3*, *HaMADS16*, *HaMADS18*, *HaMADS19*, *HaMADS20*, *HaMADS22*, and *HaMADS25*) showed higher levels at 5 DPB (Fig. 3B).

Comparing the expression levels of these genes in WT and *lpm* during floral development, the results indicated that six genes showed a reduced expression trend in *lpm* compared with WT: *HaMADS1*, *HaMADS3*, *HaMADS9*, *HaMADS11*, *HaMADS22*, and *HaMADS15* (Fig. 3B). Eight genes were down-regulated in the early stages but up-regulated later, including *HaMADS7*, *HaMADS12*, *HaMADS18*, *HaMADS19*, *HaMADS21*, *HaMADS25*, *HaMADS26*, and *HaMADS27* (Fig. 3B). Only one gene,

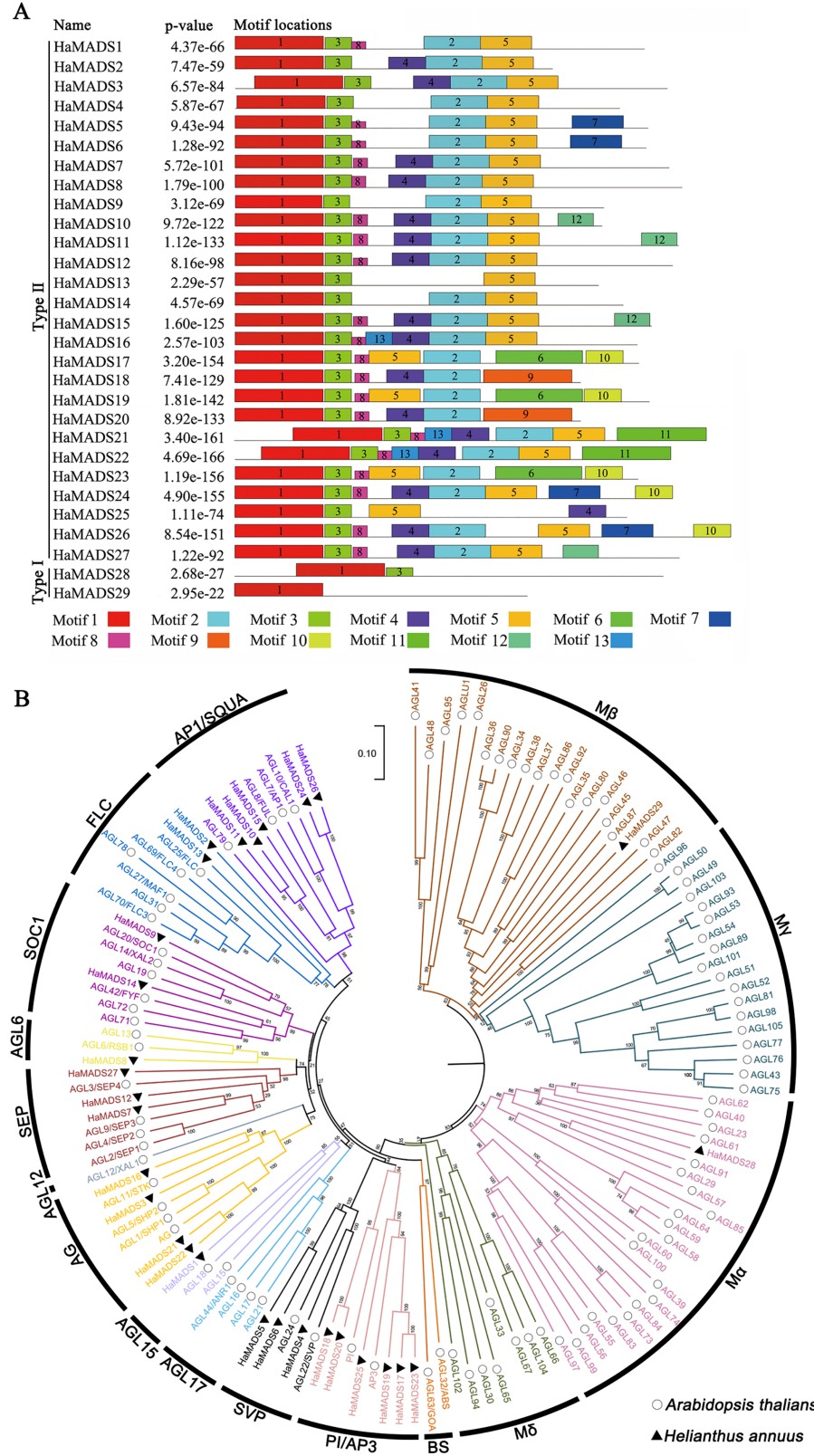

**Figure 2 Analysis of conserved motifs of MADS-box protein in sunflower and construction of phylogenetic tree.** (A) Conserved motifs of MADS-box proteins in sunflower. The conserved motifs

**Figure 2** (continued)
were visualized *via* MEME online website, and *P*-value was calculated referring to *Tanaka, Bailey & Keich, 2014*. (B) Phylogenetic tree of MADS-box genes in sunflower and *A. thaliana*. Neighbor-joining tree with bootstrap replications of 1,000, the branch length represents sequence similarity.

*HaMADS18*, was up-regulation at all stages in *lpm*. In addition, four genes, *HaMADS7*, *HaMADS8*, *HaMADS20*, and *HaMADS21*, showed a steadily high level at all stages in *lpm* (Fig. 3B).

Notably, among these genes, eighteen were found strongly inhibited in *lpm* at the stage when they presented with peak value in WT. Eleven of these genes were strongly inhibited in *lpm* at 25 DPB: *HaMADS7*, *HaMADS8*, *HaMADS9*, *HaMADS11*, *HaMADS12*, *HaMADS15*, *HaMADS21*, *HaMADS23*, *HaMADS24*, *HaMADS26*, and *HaMADS27* (Fig. 3B). Similarly, the other seven genes were inhibited in *lpm* at 5 DPB: *HaMADS1*, *HaMADS3*, *HaMADS16*, *HaMADS19*, *HaMADS20*, *HaMADS22*, and *HaMADS25* (Fig. 3B). Perhaps these genes are related to the development of abnormal flower organs, such as petal and stamen, in *lpm*.

## Identified MADS-box genes associated with petal development

To further identify the MADS-box genes related to the formation of abnormal floral organs in *lpm*, the difference of their expression in floral organs of disc floret were analyzed between WT and *lpm*. The results showed that 12 genes were involved in the development of floral organs in ray-like floret of *lpm* (*HaMADS3*, *HaMADS7*, *HaMADS8*, *HaMADS11*, *HaMADS15*, *HaMADS16*, *HaMADS18*, *HaMADS19*, *HaMADS20*, *HaMADS21*, *HaMADS24*, and *HaMADS26*) due to their regular expression discrepancy in floral organs between WT and *lpm* (Figs. 4 and S2). Among these genes, only five displayed expression variation in petals between WT and *lpm*: *HaMADS21*, *HaMADS3*, and *HaMADS11*, which were suppressed seriously in *lpm* petals from 5 DPB, and *HaMADS7* and *HaMADS8*, which were significantly up-regulated (Fig. 4). And these five genes were considered to be associated with the formation of abnormal petal in disc floret of *lpm*.

Taken together, we found that *HaMADS3* was strongly inhibited not only in various organs, including bract, petal, and pistil, but also in the whole process of flower development in *lpm*. *HaMADS7* was significantly inhibited in the early stage (25 DPB) and became gradually up-regulated as the flower developed, when compared with WT (Figs. 3B, S2 and S3). However, comparatively, it was strongly up-regulated in all floral organs of *lpm*, particularly in petals (Figs. 4, S2 and S3). The expression of *HaMADS8* exhibited strong stage-specificity in WT, with the most optimal value at 25 DPB followed by 5 DPB, but stage-specificity was weakened in *lpm* (Figs. 3B, S2 and S3). The expression of *HaMADS8* was significantly higher in *lpm* than in WT, although no obvious difference was observed among the bract, petal, and pistil in WT, nor in *lpm* (Figs. 4, S2 and S3). *HaMADS11* was strongly inhibited from 25 DPB to 5 DPB in *lpm*, and then slightly up-regulated compared with WT (Figs. 3B, S2 and S3). Its expression significantly decreased from 5 DPB to blooming only in the petal and pistil, but not in the bract of *lpm* plants (Figs. 4, S2 and S3). The expression of *HaMADS21* was repressed originally at 25

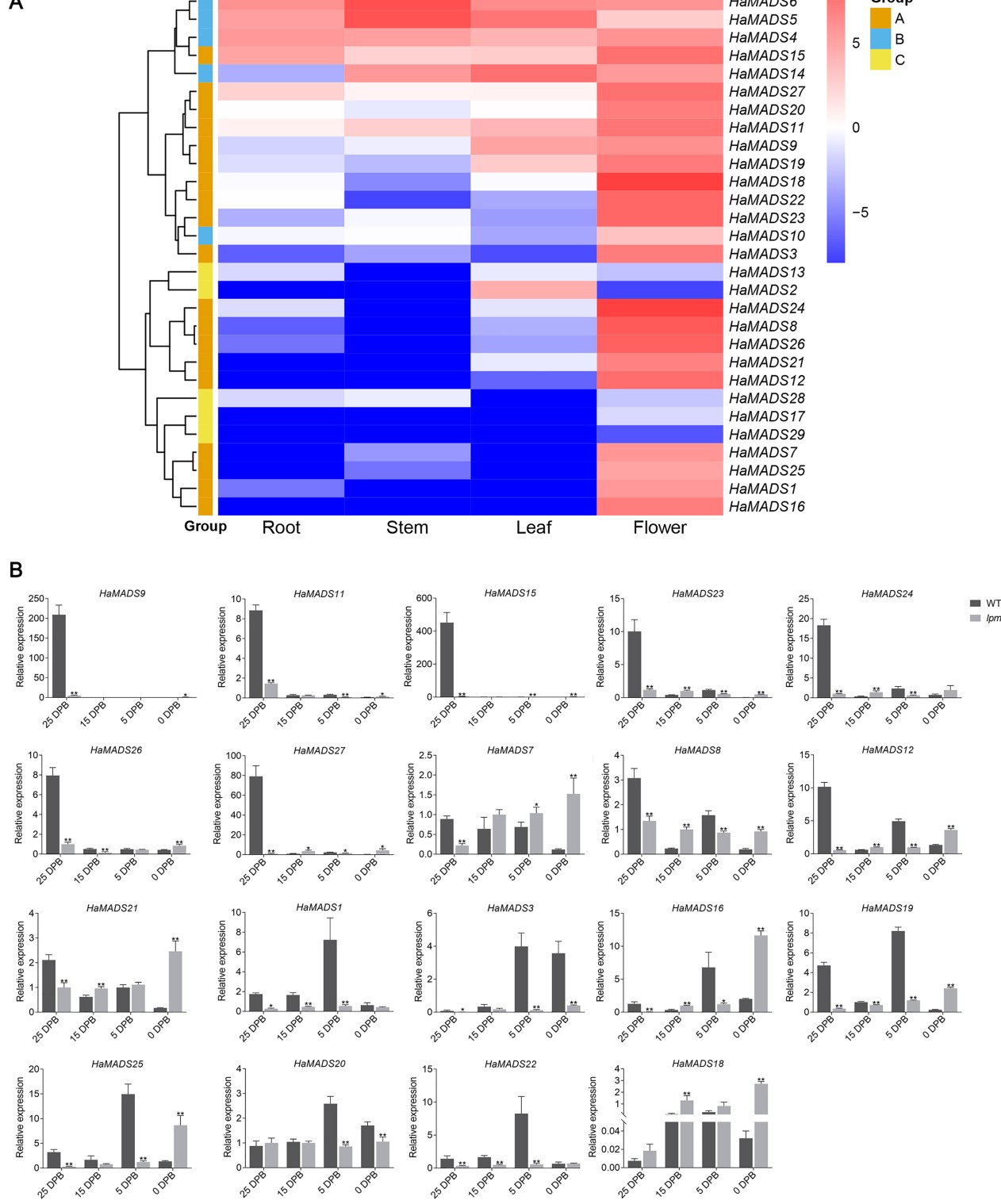

**Figure 3 Expression analysis of MADS-box genes.** (A) Tissue expression pattern of MADS-box genes in sunflower based on RNA-seq data. Color change from red to blue represents a change of expression level from high to low. Group A: the genes with highest expression level in flower. Group B: the genes expressed in all tissues. Group C: the genes hardly expressed in flower. (B) Expression pattern of MADX-box genes in different stages of flower development. Note: Each column represents mean ± SEM of three technical. An asterisk (*) represents $P < 0.05$ and two asterisks (**) represent $P < 0.01$ by one-way ANOVA analysis.

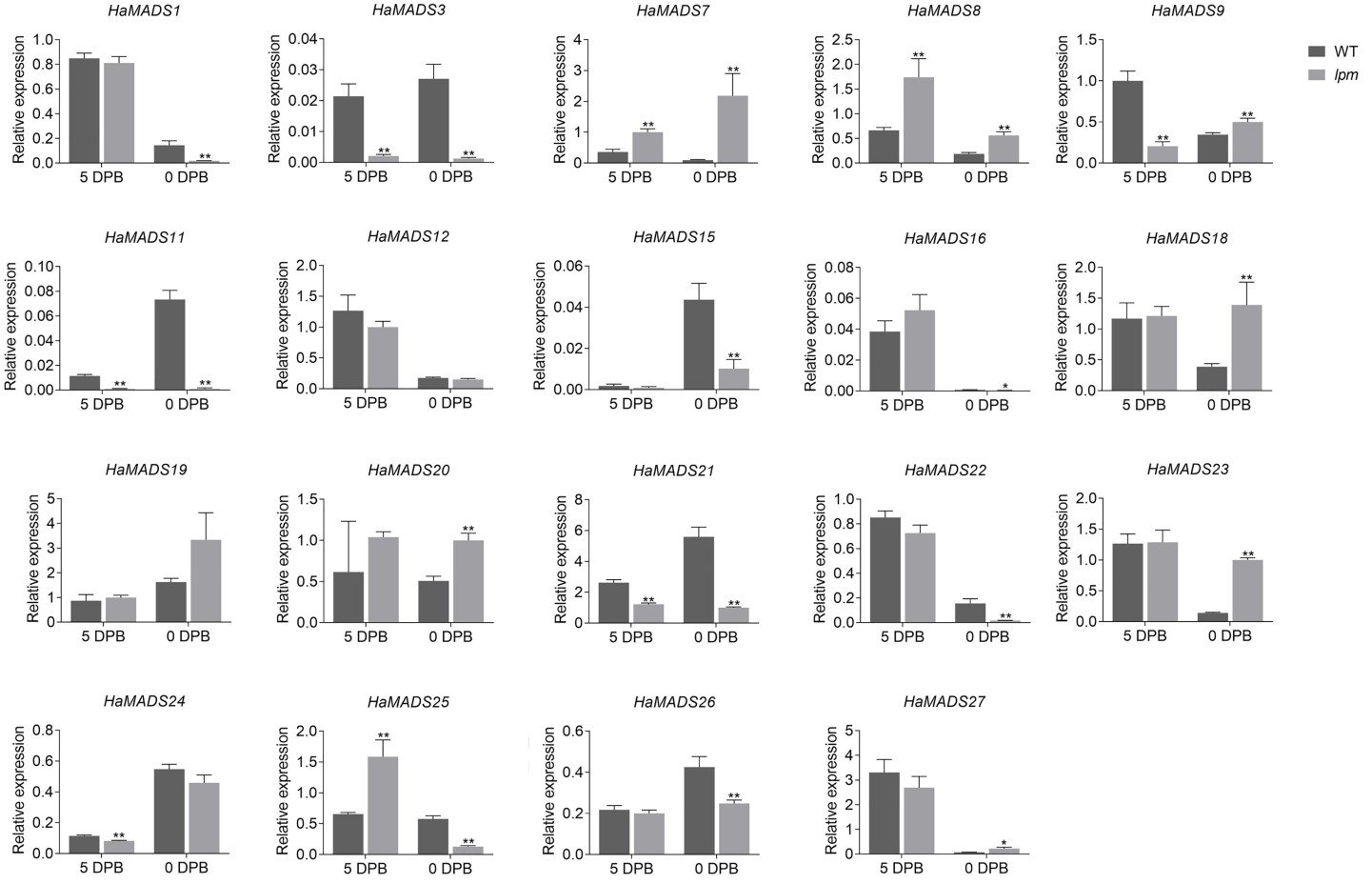

**Figure 4 Expression of MADS-box genes in petals at 5 DPB and 0 DPB.** Note: Each column represents mean ± SEM of three technical. An asterisk (*) represents $P < 0.05$ and two asterisks (**) represent $P < 0.01$ by one-way ANOVA analysis.

DPB, but up-regulated from 15 DPB to blooming in *lpm* when compared with WT (Figs. 3B, S2 and S3). Its expression was inhibited in all floral organs except for bract in *lpm* (Figs. 4, S2 and S3). As the spatiotemporal expression results of *HaMADS21* were contradictory, it was excluded for its contribution to the formation of abnormal floral organs of *lpm*. Therefore, *HaMADS3*, *HaMADS7*, *HaMADS8*, and *HaMADS11* were considered to play important roles during the development of floral organs, such as petals.

In order to further identify the effects of *HaMADS3*, *HaMADS7*, *HaMADS8*, and *HaMADS11* on the elongation and symmetry determinacy of petals, their expression levels were detected at various stages in ray and disc floret petals of WT. The results showed that *HaMADS3* was strongly inhibited in the ray floret petals of WT (Fig. 5), similar to the ray-like floret petals in *lpm* (Fig. 4). *HaMADS7* and *HaMADS8* were significantly up-regulated in ray floret petals compared to disc floret petals in WT and showed a similar up-regulation trend in the ray-like floret petals of *lpm* (Figs. 4 and 5). It could be deduced that the lower expression of *HaMADS3* and higher expression of *HaMADS7* and *HaMADS8* coordinately contributed to the formation of long and zygomorphic petals. On the other hand, the other gene *HaMADS11* presented with up-regulation in ray floret

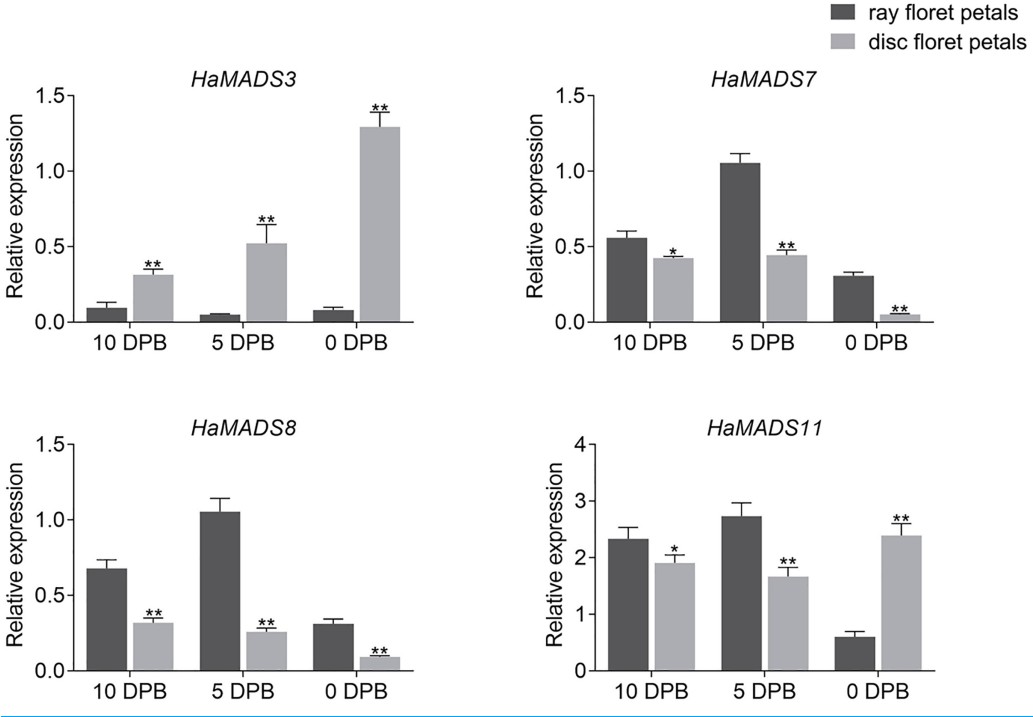

**Figure 5 Expression of MADS-box genes in disc and ray floret petal of WT plants at different stages.**
Note: Each column represents mean ± SEM of three technical. An asterisk (*) represents $P < 0.05$ and two asterisks (**) represent $P < 0.01$ by one-way ANOVA analysis.

petals when compared with disc florets in WT (Fig. 5), but was inhibited in ray-like floret petals in *lpm* (Fig. 4). For the logically conflicting exhibition of *HaMADS11* in WT and *lpm*, only *HaMADS3*, *HaMADS7*, and *HaMADS8* were considered to close association with the formation of long and zygomorphic petals in *lpm*.

Subsequently, the expression levels of *HaMADS3*, *HaMADS7*, and *HaMADS8* were investigated in floret petals within the 1st (ray floret), 5th, 15th, and 19th parastichy in WT and *lpm* plants, and the correlation coefficients between the expression levels of the three MADS-box genes and petal length were analyzed. The results showed that the expression levels of *HaMADS7* and *HaMADS8* were significantly higher in the petals of ray-like florets from *lpm* and ray florets from WT compared to disc florets from WT. The opposite expression of *HaMADS3* was detected (Fig. 6). Further correlation analysis suggested that the correlation coefficient between petal length and the expression levels of *HaMADS3*, *HaMADS7*, and *HaMADS8* was −0.99, 0.96, and 0.97, respectively, suggesting a positive correlation between petal length and expression level of *HaMADS7* and *HaMADS8*, and a negative correlation between petal length and expression level of *HaMADS3* (Table 2). These results indicated the involvement of *HaMADS3*, *HaMADS7*, and *HaMADS8* in petal prolongation and floral symmetry establishment in sunflower.

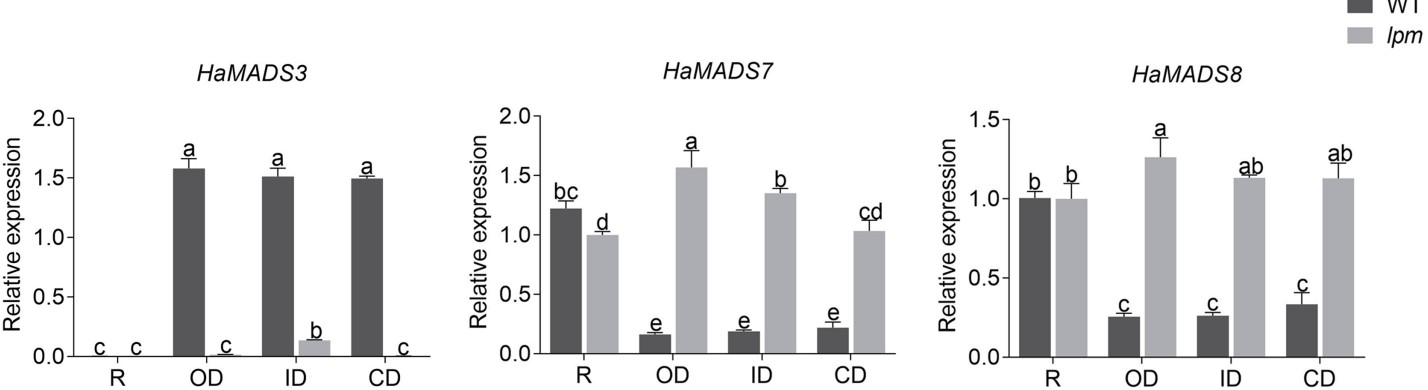

**Figure 6 Expression of MADS-box genes in petals of WT and *lpm* plants on different positions.** R: ray floret (the 1st parastichy of floret). OD: outer disc floret (the 5th parastichy of floret). ID: intermediate disc floret (the 15th parastichy of floret), CD: central disc floret (the 19th parastichy of floret). Different lowercase letters represent significant differences (P < 0.05 by one-way ANOVA analysis and Tukey multiple comparison).

**Table 2 Correlation analysis between expression levels of *HaMADS3*, *HaMADS7*, and *HaMADS8* and petal length.**

| Phenotype | Genes | Correlation coefficient | *P*-value |
|---|---|---|---|
| Petal length | *HaMADS3* | −0.99 | 1.78E-06 |
| | *HaMADS7* | 0.96 | 0.000155663 |
| | *HaMADS8* | 0.97 | 5.68E-05 |

### Interacting analysis between MADS-box genes and *HaCYC2c*

A previous study (*He et al., 2022*) found that the abnormally higher expression of *HaCYC2c* (LOC110936862) could result in the symmetrical transformation of disc floret from actinomorphic to zygomorphic in *lpm* sunflower, *via* recognizing the *cis*-elements in the promoter region of *HaNDUA2*. To further understand the interaction between these three MADS-box genes and *HaCYC2c* in regulating long and zygomorphic petal formation, their interaction relationship was analyzed. The results indicated that 14 *cis*-acting elements existed in the promoter region (−2,000 bp) of *HaCYC2c* as the potential transcriptional recognition binding sites for MADS-box transcription factors (Fig. 7 and Table 3). However, no HaCYC2c binding *cis*-acting element was found within the promoter regions (−2,000 bp) of *HaMADS3*, *HaMADS7*, or *HaMADS8* (Fig. 7 and Table 3). These results implied that *HaMADS3*, *HaMADS7*, and *HaMADS8* may regulate the expression activity of *HaCYC2c*, resulting in petal prolongation and symmetry transformation from actinomorphic to zygomorphic in *lpm* sunflower.

## DISCUSSION

The flower is an important reproductive part of plants, and its morphology could affect a plant's ability to attract pollinating insects. Symmetrical morphogenesis, one factor that affects flower morphology, may alter a plant's attraction to pollinating insects and self-reproduction (*Moyroud & Glover, 2017*; *Nakagawa, Kitazawa & Fujimoto, 2020*).

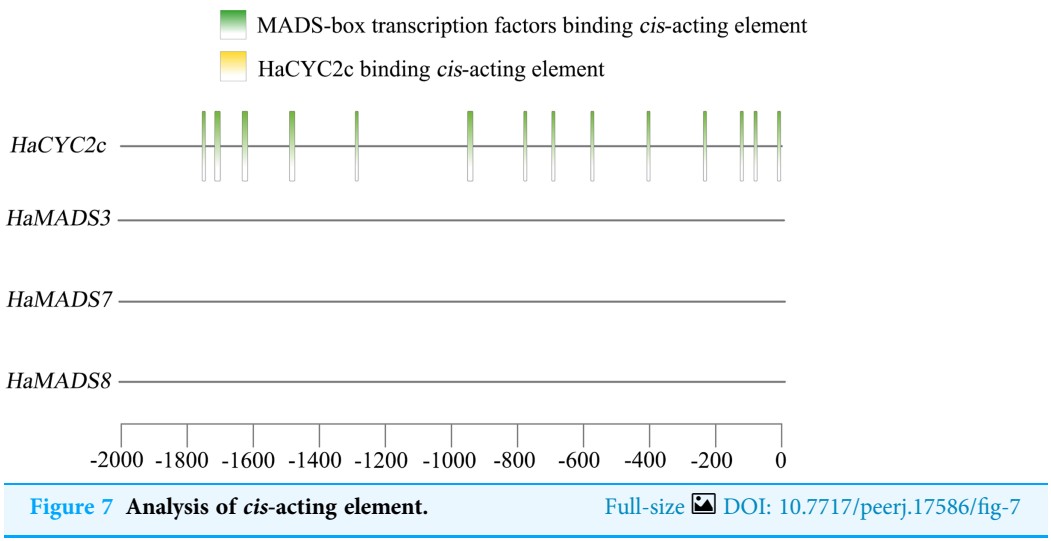

Figure 7 Analysis of *cis*-acting element.               

Table 3 Analysis of *cis*-acting elements.

| Gene name | Start | End | Predicted sequence | Annotation |
|---|---|---|---|---|
| *HaCYC2c* | −84 | −74 | CCTAAAAAGGA | MADS-box transcription factors binding element |
| *HaCYC2c* | −1,754 | −1,744 | CCGAATACAGA | MADS-box transcription factors binding element |
| *HaCYC2c* | −696 | −686 | ACTAATTGGGC | MADS-box transcription factors binding element |
| *HaCYC2c* | −1,633 | −1,616 | ATTACGGAATTTGTCAAA | MADS-box transcription factors binding element |
| *HaCYC2c* | −578 | −568 | CACAAATAGGT | MADS-box transcription factors binding element |
| *HaCYC2c* | −408 | −398 | CCTTATTTAAT | MADS-box transcription factors binding element |
| *HaCYC2c* | −781 | −771 | CACTTTTTTGG | MADS-box transcription factors binding element |
| *HaCYC2c* | −14 | −4 | GCTTTATTTGG | MADS-box transcription factors binding element |
| *HaCYC2c* | −951 | −934 | CTTTCCATACATAGCAAC | MADS-box transcription factors binding element |
| *HaCYC2c* | −1,490 | −1,473 | TTTACTAAAATTAGAAGT | MADS-box transcription factors binding element |
| *HaCYC2c* | −126 | −116 | TATACTTTTGT | MADS-box transcription factors binding element |
| *HaCYC2c* | −237 | −227 | CCTATTTTTAT | MADS-box transcription factors binding element |
| *HaCYC2c* | −1,291 | −1,281 | CACATATTTGT | MADS-box transcription factors binding element |
| *HaCYC2c* | −1,713 | −1,703 | TCTATTATTGT | MADS-box transcription factors binding element |
| *HaMADS3* | No HaCYC2c binding element | | | |
| *HaMADS7* | No HaCYC2c binding element | | | |
| *HaMADS8* | No HaCYC2c binding element | | | |

The sunflower is a typical member of Asteraceae and possesses a capitulum comprised of several whorls of actinomorphic disc florets and a whorl of zygomorphic ray florets (*Elomaa, Zhao & Zhang, 2018*). Infertile ray florets have a strong attraction to recruiting pollinators and assist the fertile disc florets in producing offspring. Ray florets and disc florets can synergistically maintain the reproduction of sunflower. In order to implement the synergistical action of the two types of florets in sunflower, floral organ primordia can develop differentially into various floral organs, thus producing morphological differences

between ray and disc florets, such as petal length and the floret symmetry (*Celedón-Neghme, Gonzáles & Gianoli, 2007*). According to previous reports, as the major members of ABCDE model, MADS-box genes affect floral symmetry by regulating the development of floral organs during this process (*Sasaki et al., 2014*). Among these genes, A-class genes functioned in the first and second whorl, B-class genes functioned in the second and third whorl, C-class genes functioned in the third and fourth whorl, and E-class genes functioned in all four whorls (*Theißen, Melzer & Rümpler, 2016*; *Patil et al., 2023*). In this study, 29 MADS-box genes were identified, and 19 members were found with flower-specific expression. After further analysis, only three genes were identified to be closely related to petal development and symmetry determinacy: *HaMADS3*, *HaMADS7*, and *HaMADS8*.

In general, C-class MADS-box genes acted in the formation of stamen and pistil, and functioned in inhibiting petal development in multiple species (*Mizukami & Ma, 1992*; *Zhang et al., 2020*; *Li et al., 2022*). A previous report showed that the overexpression of *AG* could result in shorter and narrower petals compared with WT plants in *A. thaliana* (*Mizukami & Ma, 1992*). Furthermore, C-class genes could regulate cell proliferation during the process of floral organ development (*Noor et al., 2014*). In *Petunia hybrida*, the inhibition of the C-class genes *PETUNIA MADS-BOX GENE3* (*Pmads3*) and *FLORAL BINDING PROTEIN6* (*FBP6*) could increase the cell number of stamen, suggesting the inhibitory role of C-class genes in the cell proliferation of floral organs (*Noor et al., 2014*). As a member of C-class genes, perhaps *HaMADS3* has similar inhibitory effects on cell proliferation in sunflower, because the cell number of ray-like floret petals in *lpm* was far greater than that in disc florets from WT (Fig. 1G). Meanwhile, the expression of *HaMADS3* was strongly inhibited in the ray-like floret petals of *lpm* and was negatively correlated with petal length (Figs. 4, 6 and Table 2). These results suggested that the strong inhibition of *HaMADS3* relieved the inhibition of cell proliferation and promoted the petal prolongation of ray-like florets in *lpm*. In addition, C-class genes could determine the identity of stamen and pistil, as well as regulate their development (*Zhang et al., 2020*). In this study, the degeneration of stamens was closely accompanied by prolonged petals in the ray-like florets of *lpm*. Perhaps the acute expression inhibition of *HaMADS3* contributed to the degeneration of stamen in *lpm*; however, the exact mechanism needs further investigation.

E-class genes are involved in the development regulation of all floral organs (*Pu & Xu, 2022*). The E-class genes *GRCD4* and *GRCD5* were reported to effectively promote the extension of petals in *G. hybrida* (*Zhang et al., 2017*). As a homologous gene of these genes, *HaMADS7* was intensely up-regulated in ray-like floret petals and its expression level was positively correlated with petal length in *lpm*, indicating that its higher expression could effectively accelerate petal prolongation. It was reported that AGAMOUS-LIKE6 (AGL6) clade genes could determine the identity of floral organ and regulate their development (*Dreni & Zhang, 2016*). Tomato *SlAGL6* transcription deletion seriously inhibited petal development, leading to the formation of smaller petals (*Yu et al., 2017*), suggesting the promotion of *SlAGL6* on the extension of tomato petals. In this study, *HaMADS8*, as a member of AGL6, was found with steady expression levels in *lpm*. Its expression level was

significantly up-regulated in all floral organs, especially in petals, and was positively correlated with petal length. Therefore, a higher expression of *HaMADS8* was crucial for petal extension in *lpm*.

Overall, the expression alteration of *HaMADS3*, *HaMADS7*, and *HaMADS8* could impact floral organ development and floret symmetry determinacy. The high-level expression of *HaMADS7* and *HaMADS8* could promote petal development and elongation in ray-like florets of *lpm*. Simultaneously, the inhibition of *HaMADS3*, a cell division inhibitor of floral organs, could result in the rapid proliferation of petal cells for petal extension in *lpm*. These results suggested that the abnormal expression of these genes would give rise to the formation of ray-like florets in *lpm*. For WT plants, higher expression of *HaMADS3*, together with lower expression of *HaMADS7* and *HaMADS8*, could repress the rapid elongation of petal and maintain the normal morphology of disc florets. Conversely, the lower expression of *HaMADS3* and higher expression of *HaMADS7* and *HaMADS8* could effectively promote the rapid elongation of petal and result in the formation of the zygomorphic ray florets in WT. Our results suggested that *HaMADS3*, *HaMADS7*, and *HaMADS8* play an important role in the development of sunflower florets, especially in petal length and symmetry establishment.

According to previous reports, the abnormally higher expression of *HaCYC2c* could result in the symmetrical transformation of disc floret from actinomorphic to zygomorphic in *dbl* and *Chry2* sunflower, and its function deficiency would cause the symmetrical transformation of ray floret from zygomorphic to actinomorphic in *tub* ang *turf* sunflower (*Chapman et al., 2012*; *Fambrini et al., 2014*). Furthermore, it was reported that the abnormally higher expression of *HaCYC2c* could result in the symmetrical transformation of disc floret from actinomorphic to zygomorphic in *lpm* sunflower (*He et al., 2022*). Therefore, *HaCYC2c* is a critical factor in floral symmetry determinacy. In this study, some potential MADS-box binding sites were found in the promoter region of *HaCYC2c*, but no *cis*-acting element was found to be transcriptionally recognized by HaCYC2c in the promoter regions of *HaMADS3*, *HaMADS7*, and *HaMADS8*. Previous reports suggested that the C-class MADS-box transcription factor GAGA1 and E-class MADS-box transcription factor GRCD5/GRCD8 could activate CYC2 clade gene *GhCYC3* expression during ligule development in *G. hybrida*, indicating that the genes from MADS-box family and CYC2 clade could be involved in the determinacy of flower symmetry (*Yu, 2020*; *Zhao et al., 2020*). In *Chrysanthemum*, B-class MADS-box gene *CDM19* may positively regulate the activity of *CmCYC2c* and *CmCYC2d* (*Chai et al., 2023*). It could be deduced that the three MADS-box genes may regulate the expression of *HaCYC2c* to contribute to petal prolongation and symmetry conversion of the disc florets of *lpm*. These results suggested that *HaMADS3*, *HaMADS7*, and *HaMADS8* could play important roles in petal prolongation and floral symmetry establishment in sunflower and are closely related to *HaCYC2c*. To further confirm the functions of these MADS-box genes on flower development, especially in petal development and floral symmetry establishment, we plan to construct transgenic lines of overexpressing or silencing these MADS-box genes in further experiments. Furthermore, yeast one-hybrid, dual-luciferase reporter, and electrophoretic mobility shift assays will be performed to explore their relationship with

*HaCYC2c* during the processes of petal development and floral symmetry establishment in sunflower.

## CONCLUSION

In conclusion, the expression levels of *HaMADS3* were higher in ray-like floret petals of *lpm* compared to disc floret petals of WT, and was higher in disc floret petals compared to ray floret petals in WT plants, while *HaMADS7* and *HaMADS8* presented with opposite expression patterns. These results indicated that high levels of *HaMADS3*, along with the low levels of *HaMADS7* and *HaMADS8*, promoted petal prolongation and zygomorphic establishment of ray florets in sunflower. Correspondingly, the converse expression of these genes could repress petal extension and actinomorphic establishment of disc florets. These three genes play important role in maintaining the morphology of disc and ray florets in sunflower, and function synergistically to maintain the balance between attracting pollinators and producing offspring of sunflowers.

### Funding

This work was supported by the Natural Science Fund of Sichuan (No. 2022NSFSC0163), Natural Science Fund of Sichuan (No. 2023NSFSC0226), the Doctoral Scientific Research Startup Found of China West Normal University (No. 18Q039), and the Innovation Team Funds of China West Normal University (KCXTD2023-7). The funders had no role in study design, data collection and analysis, decision to publish, or preparation of the manuscript.

### Grant Disclosures

The following grant information was disclosed by the authors:
Natural Science Fund of Sichuan: 2022NSFSC0163.
Natural Science Fund of Sichuan: 2023NSFSC0226.
Doctoral Scientific Research Startup Found of China West Normal University: 18Q039.
Innovation Team Funds of China West Normal University: KCXTD2023-7.

### Competing Interests

The authors declare that they have no competing interests.

### Author Contributions

- Qian Wang conceived and designed the experiments, performed the experiments, analyzed the data, prepared figures and/or tables, and approved the final draft.
- Zhou Su conceived and designed the experiments, performed the experiments, prepared figures and/or tables, and approved the final draft.
- Jing Chen performed the experiments, prepared figures and/or tables, and approved the final draft.
- Weiying Chen performed the experiments, prepared figures and/or tables, and approved the final draft.

- Zhuoyuan He analyzed the data, authored or reviewed drafts of the article, and approved the final draft.
- Shuhong Wei analyzed the data, authored or reviewed drafts of the article, and approved the final draft.
- Jun Yang analyzed the data, authored or reviewed drafts of the article, and approved the final draft.
- Jian Zou conceived and designed the experiments, analyzed the data, authored or reviewed drafts of the article, and approved the final draft.

## Data Availability

The raw data of RNA-sequencing are available at SRA: PRJNA1083261.

The raw measurements are available in the Supplemental Tables.

## Supplemental Information

Supplemental information for this article can be found online at http://dx.doi.org/10.7717/peerj.17586#supplemental-information.

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
