# Peer review of "HaMADS3, HaMADS7, and HaMADS8 are involved in petal prolongation and floret symmetry establishment in sunflower (Helianthus annuus L.)"

_PeerJ, doi:10.7717/peerj.17586_

## Round 0.1 · original submission · Major Revisions

Dear Authors

The manuscript cannot be accepted for publication in its current form. It needs a major revision to be reconsidered for publication. The authors are invited to revise the paper considering all the suggestions made by the reviewers. Please note that requested changes are required for publication.

With Thanks

Reviewer 1 ·

Basic reporting

The manuscript is scientifically sound and study some sunflower TFs associated with petal development. The results suggested that HaMADS3, HaMADS7 and HaMADS8 TFs could regulate the petal development. This paper is well written and arrives to novel knowledge on a non-model relevant oil crop.

Experimental design

Methods are consistent and well described.
Nevertheless, the authors mention MIQE guidelines but they fail in adding a second reference gene (RG according to MIQE) to provide with additional standard non variable RG.
I also ask the authors if they use EF1alfa due to the one published in sunflower for gene expression analysis (Fernandez et al 2011: DOI10.1007/s00299-010-0944-3). If this is the case, please cite and/or justify.

Validity of the findings

No comments.

Additional comments

No comments.

Reviewer 2 ·

Basic reporting

This manuscript by Wang et al. reported three MADS transcription factors involved in petal prolongation and floret symmetry establishment in sunflower. This study employed bioinformatics to identify candidate MADS genes and associated their gene expression levels in WT and ppm mutant with phenotypes. The manuscript is well-written with a good logical flow. However, there are points to be addressed before it can be accepted.

Experimental design

Well-designed.

Validity of the findings

One major issue here is that only phylogenetic gene expression analyses cannot be used to confirm their involvement in petal prolongation and floret symmetry establishment. To use the tile as it is, the authors need to show functional characterization of those MADSs.

Additional comments

1. the manuscript should be proof-read by a fluent speaker. e.g., line 198, Cis-acting element prediction
2. In abstract, lack of cis-acting element prediction result.
3. The authors should make transcriptome database available publicly (line 151)
4. The authors need huge improvement on M&M section
4.1 Line 140, provide year
4.2 line 145, what is DPB
4.3 Line 162, 195, 201, 202 provide appropriate reference, not the websites
4.4 revise details of maker, country in a correct format, Brand, city, country
4.5 much better and newer version of MEGA is available, why the authors still use MEGA 6.0
5. no scale bar available in Fig. 1
6. Fig. 3, how did the heat map was constructed. The authors mentioned that red is up regulation by comparing to what organ? e.g. HaMADS6, all tissues are in red? what does this mean.
7. what is the limitation of this study? what can be further done to confirm the functions of these MADS, can be discussed.

Reviewer 3 ·

Basic reporting

The manuscript is clear, unambiguous and technically correct. Introduction, background issues and references cited are sufficiently reported. The article structure is acceptable. With respect to figure 1, the panels B, C and E are similar to those published previously by the same authors (Plant Cell Reports 41, pages 1025–1041 (2022)). Moreover, it should also be noted that panel E is not immediately readable and its contents should be better explicated (for example, rewrite the legend). Why the block WT and lpm are repeated tree times from the top down? Raw data have been checked and they are consistent.

Experimental design

Aims and scope of the Journal have been respected and the research questions are clear. Moreover, the Methods are described with sufficient detail & information. However, the fact that “These lpm plants could be divided into five types, including type V, IV, III, II and I (lines 131-136)” may suggest that we are dealing with plant material that is heterogeneous from a genetic point of view, since the presence of plants with intermediate phenotype might suggest non-homozygous conditions for the mutation (e.g., see Ann. appl. Biol. 2003, 143:341-347). In this regard, has the genetic control of the lpm character been determined?

Validity of the findings

In this manuscript Wang et al screened 29 MADS gene in sunflower during inflorescence development using two genotypes: wild type and the lpm mutant characterized by chrysanthemoides phenotype (without disc actinomorphic floret with a small size of corolla). This type of mutant have been studied previously by other authors but no data were available about the MADS gene activity in the florets morphogenesis. The key result reported in the manuscript is the involvement of HaMADS3, HaMADS7 and HaMADS8 genes in the regulation of petal elongation. Importantly, Wang and co-workers base their conclusions mainly by presenting gene expression data by q-PCR. The authors analyzed gene expression in different types of flowers, in isolated flower organs and samples were collected at different stages of development. However, basing major conclusions about the role of selected genes on the basis only of (quantitative and temporal) gene levels has obvious limitations. In my opinion, the authors should be very cautious in establishing specific roles of the genes they selected with respect to the determination of corolla symmetry of disc flowers; in fact, some important questions are still open: For example, where are the cellular domain in the floret primordia of each specific gene, selected? What is the phenotype of sunflower inflorescence if HaMADS3 and/or HaMADS7 are silenced? What could be the roles that genes such as HaMDS3 or HaMADS8 play on flower organs other than the corolla such as the pistil and bracts (see fig. 4). In other sunflower mutants with lpm-like phenotype, is the expression pattern of HaMADS3,7,8 genes confirmed or not?
The characterization of the expression pattern in WT and lpm mutants of MADS genes as addressed by Wang et al, has its own interest but the data collected per se are not sufficient to delineate specific roles for each gene analysed. Moreover, on sunflower mutants with chrysanthemoides phenotype, some previously obtained literature findings are omitted from the discussion (e.g., genesis 52:315–327; 2014).

Reviewer 4 ·

Excellent Review

This review has been rated excellent by staff (in the top 15% of reviews)
EDITOR COMMENT
I want to express my deep gratitude for your detailed review, which will be very helpful during the evaluation process of this manuscript. Your help is so much appreciated.

Basic reporting

The abstract does not provide a good outline for the study. It is not clear if gene expression was compared between ray and disc floret or between wild type and mutant sunflowers. I think the methodology needs to be more clearly outlined in the abstract so the results can be properly interpreted.

The introduction is a little bit scattered and too broad; the wide view of the topics causes the introduction to be vague, difficult to follow and unclear on the knowledge gaps addressed by the study. I recommend that the Introduction be rewritten. I think the first paragraph can remain the first paragraph but the terms actinomorphic and zygomorphic need to be described and defined. Then I think this needs to be followed by a paragraph introducing Asteraceae floral structure, highlighting the unique compressed inflorescence structure and the different flower forms on the head. In this paragraph it would be good to highlight what is unique to sunflower compared to other Asteraceae species and to highlight why it is important to study these mechanisms in this species. Then write a paragraph outlining what is known about the genetic regulation of floral development and the ABCDE model in Arabidopsis. Then compare this to what is known in Asteraceae and highlight differences or where there is information lacking. Next describe the mutant being used in this study, how it is different from the wild type and why it is useful. This should lead you into describing the specifics of this study.

The references of the study appear to be slightly out of date. I count that 65% of the references were published ten or more years ago. In particular in the introduction only six papers are less than 10 years old were cited. I recommend updating some or the references, especially in the introduction where new reviews on flower development, flower formation and the flowering model might be more relevant.

The results section would be easier to follow if restructured a little bit. Lines 271-296, should be a detailed description of the groups of expression patterns observed for these MADS genes over the developmental series and how they relate to the biology. Lines 207-304 should be rewritten to give a clear over view of the expression of these genes in the different floral tissues with a focus of which genes in the wild type and mutants are different in the petals at the two selected developmental stages. The results described from line 305-323 would be greatly aided by a summary schematic figure that shows the developmental stages across the top, lists the genes down the side and then indicates each gene’s expression for each developmental stage in WT and Mutant petals. Essentially produce a combination of Figure 1E, figure 3 and figure 4. This section could benefit from its own heading.
The discussion is good but I think maybe a detailed description of the ABCDE model at the start would help frame the discussion and put it in context.

See detailed comments in the additional comments section

Experimental design

It was not clear if the focus was floral specific MAD-box genes or a genome wide analysis but the aim and questions of comparing floral tissues between wild type and a floral mutant was clear.

It was difficult to get a handle on the experimental design and methods for a number of reasons. The paper cannot be published until all the issues with reporting experimental design and methods are resolved.

I suggest providing a table or figure summarizing the tissues sampled and the comparisons made. I also suggest adding to this figure or table the number of biological and technical reps performed for each sample. The authors state 15 plants were sampled, but it is not clear if these were pooled or individually run. There is a table listing the MIQE checklist, which is wonderful. But, the information in the table is inaccurate the line numbers referred to are not correct and not helpful since the line numbers will not be present in a final published version. I suggest writing out the information as requested in the table directly. Some of the information required in Table S3 is not actually in the materials and methods and needs to be added (see my comments directly on the table). Overall, with the methods in this current form the experiments will not be repeatable and more detail needs to be added

See detailed comments in the additional comments section

Validity of the findings

The results presented here are very interesting and the figures clearly indicate what is being discussed. However, it is very difficult to gauge the full validity of the results due to the very vague and limited materials and methods section.

See detailed comments in the additional comments section

Additional comments

4. General comments
Overall, I think this is an interesting and important study, unfortunately it can not be published in its current form. The major issue with the manuscript is the Materials and Methods section. This section is too vague and does not provide a good basis off which the experiment could be replicated or from which the results can be unambiguously interpreted. I also feel the Introduction and Results section could be rewritten to make them more specific and focused on the topic of petal elongation in Asteraceae.

5. Specific comments
Line 30: “especially on petal development” In this place in the abstract it would be good to highlight that sunflower has two different types of petals that are different in form between ray and disc florets. I think a sentence or two on sunflower floret structure would help frame the study.

Line 31-32: “and their expression in WT and long petal mutant (lpm) plants was analyzed” Providing one or two sentences introducing the mutant and its phenotype will help frame the study.

Line 53: “angiosperm, flower” There should be a “the” in front of the word “flower”

Line 57: “development originated from floral” it should be “originates” not “originated”

Line 57: “meristems could differentiate” Remove the word “could” or change it to “can”

Line 56-59: “Usually, the flower development originated from floral induction, and shoot apical meristems could differentiate into inflorescence meristems, then transform into floral meristems (FMs) and trigger the floral organogenesis.” This statement needs the support of a good up-to-date review paper of floral induction and flower formation.

Line 59-61: “During this process, flower homologous genes could promote FMs to differentiate into various floral organ primordia, and maintain their identity to achieve floral organogenesis” This statement is not clear or precise. What is a “flower homologous gene”? Gene homology generally refers to genes with similar sequences and structures. How does this relate to flowers? I think this needs to be elaborated on or clarified.

Line 67: “actinomorphic type” please add and description and definition of this terminology it will add to the understanding of this and the next paragraph.

Line 81-84: “For Chrysanthemum rhombifolium, the dorsal, the lateral and the ventral petal primordia could develop differentially, for their later fusion to form zygomorphic florets” It is strange to mention a chrysanthemum example, but not highlight that sunflower ray florets are also actinomorphic. Add this information given that the focus of the study is on the synchronous and asynchronous development of sunflower petals.

Line 94-97: “A. thaliana, the deficiency of these MADS-box genes would result in the uncertainty of floral organ identity, leading to an abnormal morphogenesis, while they could be partially recovered by genetic complementation” There needs to be some elaboration on the role of MADS-box genes in the ABCDE flowering model in Arabidopsis. What is their role in this model? This question should be answered here.

Line 124-126: “It was found that HaMADS3 (LOC110941381), HaMADS7 (LOC110865152) and HaMADS8 (LOC110894044) could regulate the establishment of two distinct symmetries in sunflower florets by adjusting their expression” I think this statement does not clearly highlight the novelty and interest of the finding according to what is highlighted in the abstract. These genes are not only differentially expressed, but directly oppositely expressed. This should be reflected in this statement.

Line 126-128: “Our findings can provide some new clues for the molecular regulatory network of floral shape and function differentiation of florets in sunflower” This statement is very weak and uninformative. The final statement of the introduction should strongly present why these findings are important and novel. Be specific about what new “clue” are presented here to add clarity to the regulatory network and why is it important to the field.

Line 131-132: “was treated by space-mutation via “Shenzhou No.4” (Wu et al., 2020)” This definitely requires some further elaboration. Mutation by space-travel is not a common method of mutation induction. A description of what a space-mutation is, is required here.

Line 135: “extreme phenotype that almost disc floret petals” Replace “that almost” with “phenotype, where most the”

Line 136: “In this experiment,” replace “this” with “our”

Line 137: “lpm plant was” change to “plants”

Line 140: “All plants were seeded in March, and collected in July and August.” Replace “seeded” with “sown”. Also provide the year the trial took place in and the GPS coordinates of the field. The growth conditions should be written out in full, provide the irrigation method, fertilization strategy, weeding strategy and the average temperature and rainfall for the region.

Line 143: “gathered to compare” Change “gathered” to “harvested”

Line 143: “their petals, from different positions of capitulum” The phrase “different positions” is not precise and cannot be replicated. State the exact positions of the florets that where harvested. Which pseudowhorl did each represent? There is also a typo there “of” should be replaced with “on the”

Line 144: “analyze the petal length in size” How was size determined, state if area or length was measured. Where on each petal were the measurements taken to keep the measurement consistent? In this current form this is not repeatable please write out the protocol in full.

Line 147: “dissection and paraffin sections” This is not complete. State the kind of microscopy performed and the type of microtome used. This is not repeatable in its current form; the full methods and materials are required.

Line 147-148: “the 5th round, the 15th round, the 19th round” What does a round refer to? In a sunflower the developmental spirals out from the center of the capitulum are usually referred to as parastichies and the set of florets opening on the same day in a concentric circle are referred to as pseudowhorls (see: https://doi.org/10.7554/eLife.80984). Please use precise terminology.

Line 149: “lower of petal,” please add the word “part” between lower and petal.

Line 151-155: “The MADS-box genes were screened from our transcriptome database (not available online) via web sites, including Genbank of National Center for Biotechnology Information (NCBI) (https://www.ncbi.nlm.nih.gov/genbank/), the European Molecular Biology Laboratory (EMBL) (https://www.embl.de/), and DNA Data Base of Japan (DDBJ) (https://www.ddbj.nig.ac.jp/index-e.html).” This line needs explanation if the transcriptomic data sets are NOT available as stated why list all the databases where they cannot be found? If these datasets are not available this might not meet PeerJ’s policy on data sharing (PeerJ - About - Journal Policies & Procedures). If the transcriptomes are proprietary then at the very least the full MADs gene sequences used in the phylogeny should be published at one of these repositories. Also, how were these genes screened? What method was used to retrieve the MADS-box like sequences? BLAST?

Line 155-158: “The protein sequences of the reported MADS-box 156 genes were aligned with candidate genes by DNAMAN 8.0 (https://dnaman.software.informer.com/), and the unreported genes were identified in this experiment.” This statement is concerning, what genes exactly were the putative MADs box genes aligned to? “candidate genes” is not informative, the accession numbers or NCBI IDs need to be provided for the reference genes used in the alignment. What does the statement “the unreported genes were identified in this study” mean? The genes have not been published or placed on NCBI or does it mean using this method the study genes were identified as MADS-box genes?

Line 161: “MADS-box genes of sunflower and A. thaliana” The actual accession numbers and gene IDs need to be provided for all of these genes from NCBI and TAIR.

Line 165-170: “The roots, stems, leaves, and flowers from WT were collected for analyzing tissue expression pattern of MADS-box genes. The disc floret materials of WT and lpm at different stages were dissected for exploring the temporal expression pattern of these genes. The disc petals, pistils, and bracts of WT and lpm were used to investigate the spatial expression patterns of these genes, and the petals from ray and disc florets in WT were applied to understand the function of candidate MADS-box genes during the process of petal development.” There seems to be quite a complex tissue sampling strategy at play here, or is it written in a convoluted manner. It is very hard to follow which samples were harvested and compared in the gene expression analysis. I recommend a table or figure in the supplementary file showing which tissues were sampled from the WT and the mutant and which tissues were compared in the gene expression analysis.

Line 171: “15 individuals” it is not clear from this material and methods section how these samples were handled. Was RNA extracted individually for each sample for each plant? Or were the samples pooled across the 15 plants? This detail is required to interpret the results. From the RNA extraction table S1 it looks like the were pooled. Is that correct? Please state correctly how the samples were managed.

Line 176: “1% agarose gel electrophoresis” The buffer used to run the gel needs to be named. And the dye used for visualization (ethidium bromide/gel red etc) needs to be listed to make the materials complete.

Line 178: “Figure S1” This file format would not open with the standard tools available to me. I would suggest loading this as a PDF. I could not evaluate this figure.

Line 194-195: “For each sample, three technical replicates were conducted” How many biological repeats were included. Thus far in the materials and methods it is difficult to understand if there were 15 biological replicates or if these were pooled? Please explain the structure of the design.

Line 195-196: “using ef1A as reference gene.” I am under the impression that general consensus for qPCR is to use the geometric mean of multiple reference genes as a standard reference. I think some reasoning and evidence is required to explain why only one reference gene is used and some evidence provided that this gene’s expression is at a moderate level of expression that can be appropriately compared to the test genes and that this gene is stably expressed with little variation across the tissues being evaluated.

Line 212: “was consisted of” This is a typo it read “consists of”

Line 216: “compared with WT.” The end of this sentence should refer to a figure so the reader can see the data supporting this statement. Just add “Figure 1D”

Line 232: “insignificant difference was” this should be changed to “insignificant differences were”

Line 247: “was tested in ray floret petals” this should be changed to “was observed in ray floret petals”

Line 251: “Based on our transcriptome database, referring to the genome database” There was no mention of a genome database in the materials and methods? Was this on the Heliagene browser or an in-house genome sequence? This needs to be fully described in the methods section.

Line 252: “were screened out” Change “were screened out” to “were identified”

Line 259-261: “The results of the phylogenetic analysis indicated that these 29 MADS-box genes were distributed in 11 clades, and the detailed 261 information was listed in Fig. 2B.” I am surprised by the MADS gene identification and phylogenetic tree. Given the highly repetitive nature of the sunflower genome I would have expected to see far more MADS-box genes from Sunflower than from Arabidopsis. But the tree represents the 29 MADS-box sunflower genes you identified and ~105 Arabidopsis genes. Is this based on the methodology? The methodology of how the sunflower MADS-box genes in this study were identified is not clear. Line 252 states “they were screened” out of the transcriptome with support from the genome. If the transcriptome database was used this needs to state. What tissues or treatments were used for the development of this transcriptome database and if the aim of this analysis was to identify tissue specific MADS-box genes. Alternatively, if this is representative of all the sunflower MADS-box genes in the sunflower genome, I think it should be pointed out here that this gene family appear highly expanded in Arabidopsis and highly reduced in the Sunflower genomes. This would be an interesting finding worth exploring.

Line 268-296: “The other 10 genes expressed with high level both in flower and other organs (Fig. 3A), or hardly expressed in flower were ignored for further investigation.” It should be noted in the results section that four genes at the top of the heat map (MADS 6, 5, 4, 15) appear to be generalists expressed in all tissues tested in this study. While fifteen of the MADS-box genes were flower specific. It needs to be stated if this is due to the methods of MADS-box gene identification from the transcriptome database or a general trend for sunflower MADS-box genes. This could be a very interesting finding if properly supported by the appropriate methodology.

Line 287: “HaMADS18, exhibited up-regulation trend at all stages” looking at the graph cited as evidence for this statement, the statement is inaccurate. HAMADS18 does not appear to be up regulated at 25DPB, it is important that statements about the results are correctly reflected in the figures supporting the statement.

Line 295-296: “Perhaps, these genes contributed to floral organ development in sunflower, such as petal and stamen.” I think there is much more that can be stated given Figure 3B. There are a set of genes that are down regulated in the wild type and up regulated in the mutant at 0 DPB and this could indicate these genes are required to be off at late stages of bloom. While 7 and 21 are interesting as the mutants and WT have invers expression patterns over the developmental stages, suggesting these genes are critical for correct petal development. These results need to be much more deeply interrogated based on the developmental stage and comparisons between WT and mutant. A hypothesis for normal functioning of these genes during floret development could be proposed based on these patterns in Wild type and mutant.

Line 305: “Summing previous results achieved in this experiment” replace this statement with “taken together”

Line 319: “spatiotemporal expressing results” this is a typo and should read “spatiotemporal expression”

Line 341 and 343: “predicted cis-acting elements were” generally cis should be italicized.

Line 345: “2000bp) of HaMADS3, HaMADS7, and HaMADS8 (Fig. 6 and Table 2).” I think this paragraph should end with a summary sentence stating what the presence and absence of these transcription factors might mean for the regulation of these genes.

Line 347: “Flower is the important reproductive system of plant” This grammar should be changed here to “The flower is an important reproductive system of plants”

Line 361: “screened out” change to “identified”

Line 364: “C-class genes” does this mean C-class MADS genes? This is unclear please clarify

Lines 364-368: “in the formation of stamen and pistil, and functioned in inhibiting petal development (Mizukami& Ma, 1992; Zhang et al., 2020; Li et al., 2022). Previous report showed that the overexpression of AG could result in shorter and narrower petals compared with WT plant (Mizukami& Ma, 1992). Furthermore, C-class genes could also regulate cell proliferation during the process of floral organ development (Noor et al., 2014).” Is this in Arabidopsis? The species needs to be stated.

Lines 372-373: “HaMADS3 was identified with similar inhibitory effect on cell proliferation in sunflower.” What data in this manuscript supports the link between cell proliferation and the MADS3 gene? I did not see this and it would be helpful to the reader to clearly indicate which figure to look at.

Line 380-381: “HaMADS3 would contribute to the degeneration of stamen in lpm” I think it is important to highlight here that in this study stamen were not investigated and this should be done in the future. It might also be worth noting this gene was highly upregulated in Pistils in this study. How does that relate to the known functions of its orthologs?

Line 394: “Combinedly,” This is not an English word. change to “Taken together” or “Overall”

Line 410-411: “Furthermore, He et al” et al needs a full stop and comma after it.

Line 414: “bound by these MADS-box transcriptional factors,” this was not directly proven that these specific MADS box genes bound these specific cis-elements that would require a yeast-1 hybrid assay (or something similar). This is an over statement. All that can be said from the results presented here is that it is possible these genes could bind the CYC promoter based on the presence of potential MADS-box binding sites.

Line 417-425: “As previous reports suggested that the E class MADS-box transcription factor GRCD5 could activate the CYC2 clade gene GhCYC3 expression during the ligule development (Zhao et al., 2020), indicating that the genes from MADS-box family and CYC2 clade could be involved in the determinacy of flower symmetry. It could be deduced that the screened three MADS-box genes may function with close relation to HaCYC2c, via altering their expression in lpm, and contribute to symmetry conversion of the disc floral of lpm. These results suggested that HaMADS3, HaMADS7, and HaMADS8 could play important role in the floral symmetry establishment and differentiation of disc florets and ray florets in sunflower, closely related to HaCYC2c” I think the finding that no cyc binding sites were found in any of the MADS gene promoters is interesting and warrants some discussion here. It indicates a directional regulation and not a feedback or feed forward loop. What is known about
this regulation in other species? Is this expected?

Line 427-432: “In conclusion, the abnormal expression of HaMADS3, HaMADS7, and HaMADS8 contributed to the elongation of ray-like floret petals and the abnormality of the reproductive system in lpm plants. These 3 genes play important role in maintaining the actinomorphic disc florets and zygomorphic ray florets in sunflower by the expression adjustment, thus function synergistically to maintain the balance between attracting pollinators and producing offspring of sunflowers” I think the conclusion could be more detailed where the regulatory system in sunflower petal formation is proposed in clear general terms based on the findings of the paper.

Figure S1: not in a format this reviewer could easily visualize. Recommend changing this to an annotated PDF

Table S2: the accession numbers of these genes need to be reflected here.

Table S3: there are some major problems with table S3, firstly the line numbers it refers to are incorrect and in some cases the information that is claimed to be in the section referred to is not present anywhere in the materials and methods section. Both the materials and methods section and this table need to be re-done to better reflect each other.

Figure 1: Panel E, it is not clear if the three rows of WT and lpm are replicated or different rounds, I suggest labeling this on the figure and mentioning it in the figure legend. But otherwise, this is a very nice figure!

Table S4: I do not think the ef1A gene should be included in this list. This is a list of the potential sunflower MADS genes. The ef1A does not fit here, I recommend removing it.
Table 1: This table would be more informative if the types were included and the gene sequences the motifs were found in were included. In its current form I am not sure I see the value of including this here.

Figure 2: I think it would be informative in figure 2A to indicate the Type I and Type II conformations on the figure. In the legend the software used to generate the figure and the statistics used to calculate the P-value must be mentioned. In figure 2B the legend needs to reflect what type of tree was constructed, what the bootstrap value was and what the length of the branches indicates.

Figure 3: In the legend of this figure, was the heat map data was based on the transcriptomes you used to identify the genes originally or if this was qPCR expression data? Indicate in the legend. What the error bars represent should be stated (STDev or SE?) and if the error is for technical or biological replicates. It is recommended that you add some indicator on the heatmap in figure 3A to show which 19 genes were carried forward for further floral specific analyses. Figure 3B will be more informative if regrouped to reflect the interesting expression patterns observed rather than in ascending order of the numbers of the genes. There are several groups of genes that have similar expression patterns and could be grouped together based on these. Collecting the graphs based on expression pattern will provide stronger biological inference and meaning to the result. I also think given these are developmental stages/timepoints it might be better to represent these as line graphs and you could put the same expression patterns on the same graphs? This might provide more insight into the different interesting patterns.

Figure 4: I think that perhaps not all of this information is required in the main manuscript as the discussion and the results is mostly focused on the petals. I recommend moving this figure as a whole to the supplementary data and then make a new figure for the manuscript that just focuses on the petal expression between the mutant and WT at the 5DPB and 0DPB time points. This will help focus the discussion to the topic at hand. The legend needs to indicate what the error bars are showing STDev or SE of technical or biological replicates?

Figure 5: The legend of this figure needs to indicate that this is wildtype only and a comparison of normal petals and disc florets. The legend should also indicate what the error bars represent standard deviation or standard error of the technical or biological replicates.

Table 2: I recommend changing the font size of this table for the column holding the recognized sequences to a smaller font so the sequences are on a single line. It will make this easier to read. The last 3 rows of this table should indicate no CYC binding site specifically. I am sure there are other transcription factor binding sites in these sequences.

Annotated reviews are not available for download in order to protect the identity of reviewers who chose to remain anonymous.

---

## Round 0.2 · Major Revisions

Dear Authors

The manuscript still needs a major revision to be reconsidered for publication. The authors are invited to revise the paper considering all the suggestions made by the reviewers.

Moreover, there are additional comments, after discussion with the relevant Section Editors:

In the abstract, the author claims, "Based on our transcriptome database, 29 MADS-box candidate genes were identified, and their roles on floral organ development, especially on the petal, were explored, via analyzing the expression levels in various tissues in WT and up plants."

How did they get these data?

The authors go from a section of collecting tissue (Morphological and histological analysis) to this: "The putative MADS-box genes were identified based on our transcriptome database from a collection of root, stem, leaf, and flower tissues (the transcriptome data related to this study were provided, including the transcript sequences (Table S1), detailed annotations (Table S2), and TPMs (Table S3) of all MADS-box genes analyzed in this study)."

How did the authors do the sequencing?

What are the metrics of these data?

How did they make the database?

How did they identify MADS-box genes?

When you do a transcriptome study, all of this information is necessary. Some of it can be found in the data submission, but you always need to provide all of this information so that if someone wants to replicate the data, they will have everything they need.

The authors provided the sequences of all the MADs-box genes, NOT their transcriptome database.

In addition, there are more comments:
1. How did you extract RNA?
2. How did you make the libraries?
3. How did you do the sequencing?
4. How much data did you get for each sample?
5. How did you analyze the data?
6. The complete transcripts submitted to SRA or similar (i.e., REAL raw data) – that includes everything from the transcriptome sequencing project.
7. The “database,” which is likely an assembly?
8. The methods for identifying these sequences – did you use a MADS-box sequence to blast, and if so, how? Looks like you used MADS-box genes in A. thaliana and H. annuus based on information in Supplemental Table 5. That should be in the methods and what blast program was used.
9. Everything else that was done in the transcriptome analysis.
OR did you pull these data out of public databases?

Authors need to look at a typical transcriptome paper and edit accordingly. In addition, there are still problems with the manuscript in many places. For example, Incomplete sentence: "While the type I plant possessed the extreme phenotype with prolongated petals and degenerated stamens in the almost disc floret (He et al., 2022).

Please note that the requested changes are required for publication.

With Thanks

Reviewer 2 ·

Basic reporting

The authors have significantly improved the manuscript in revised form. All responses to my comments have been well answered. Therefore, it can now be accepted.

Experimental design

The authors have significantly improved the manuscript in revised form. All responses to my comments have been well answered. Therefore, it can now be accepted.

Validity of the findings

The authors have significantly improved the manuscript in revised form. All responses to my comments have been well answered. Therefore, it can now be accepted.

Additional comments

The authors have significantly improved the manuscript in revised form. All responses to my comments have been well answered. Therefore, it can now be accepted.

Reviewer 4 ·

Basic reporting

Overall, this version of the manuscript is much improved. The article flows well and the Materials and methods now make it possible to fully follow the experimental design. The Introduction has better references and the information helps provide a good background for the manuscript. The figures, legends and tables are useful and helpful. The results are well described and the aim and objectives clear.

Experimental design

The materials and methods is vastly improved from the last draft and is clearly understandable and the experiments should be repeatable. The aims and objectives are clearly stated. The manuscript represent important findings that will be of interest to many Asteraceae researchers.

I still have one issue with the manuscript and this is the description of where on the sunflower head the florets where sampled. The authors state in Lines 175, 196 and 432 " in the 1st (ray floret), 5th, 15th, and 19th pseudowhorl" But this does not make sense.– I am still not sure the terminology is correct. In my experience growing sunflower a plant only flowers for 3-5 days and each day one pseudowhorl will open. That means generally plants only have 3-5 pseudowhorls on a head…do you mean rows of florets within the pseudowhorl? I can’t believe a sunflower can have 19 pseudowhols? I think this is important to get right because it is very hard to visualize how the sampling was done with the current description.

Validity of the findings

This research represents some interesting and novel findings answering questions of how floret development is regulated in the sunflower head to maintain disc and ray florets. A large amount of work has been done to show the expression patterns of the different MADS genes. This study opens many other interesting questions and is the starting point to mapping the regulatory pathways underlying this important biological process.

Additional comments

I also noted a couple of typos and a few grammar issues I assume the copy editors can correct.

Detailed comments:
Line 169 – Typo, there is a double degree sign before the C
Line 175 and line 196 and line 432 – I am still not sure the terminology is correct. In my experience growing sunflower a plant only flowers for 3-5 days and each day one pseudowhorl will open. That means generally plants only have 3-5 pseudowhorls on a head…do you mean rows of florets with in the pseudowhorl? I can’t believe a sunflower can have 19 pseudowhols?
Line 349 – Typo. There is a full stop and a comma at the end of the line after (Table S3)

---

## Round 0.3 · Minor Revisions

Dear Authors,

Significant concerns exist about the manuscript's grammar, usage, and overall readability. Therefore, we request that you revise the text to fix the grammatical errors and improve its readability. We suggest you have a fluent, English-language speaker thoroughly copyedit your manuscript for language usage, spelling, and grammar. If you do not know anyone who can do this, PeerJ can provide language editing services.

With Thanks

**Language Note:** The Academic Editor has identified that the English language must be improved. PeerJ can provide language editing services - please contact us at [email protected] for pricing (be sure to provide your manuscript number and title). Alternatively, you should make your own arrangements to improve the language quality and provide details in your response letter. – PeerJ Staff

---

## Round 0.4 · accepted · Accept

Dear Authors,

I am pleased to inform you that the manuscript has improved after the last revision and can be accepted for publication.

Congratulations on accepting your manuscript, and thank you for your interest in submitting your work to PeerJ.

With Thanks